# Biodistribution of RNA Vaccines and of Their Products: Evidence from Human and Animal Studies

**DOI:** 10.3390/biomedicines12010059

**Published:** 2023-12-26

**Authors:** Ildus Pateev, Kristina Seregina, Roman Ivanov, Vasiliy Reshetnikov

**Affiliations:** 1Translational Medicine Research Center, Sirius University of Science and Technology, 354340 Sochi, Russia; pateev.ii@learn.siriusuniversity.ru (I.P.);; 2Institute of Cytology and Genetics, Siberian Branch of Russian Academy of Sciences, 630090 Novosibirsk, Russia

**Keywords:** RNA vaccine, biodistribution, biosafety, lipid nanoparticles, blood–brain barrier, blood–placental barrier

## Abstract

Explosive developments in mRNA vaccine technology in the last decade have made it possible to achieve great success in clinical trials of mRNA vaccines to prevent infectious diseases and develop cancer treatments and mRNA-based gene therapy products. The approval of the mRNA-1273 and BNT162b2 mRNA vaccines against SARS-CoV-2 by the U.S. Food and Drug Administration has led to mass vaccination (with mRNA vaccines) of several hundred million people around the world, including children. Despite its effectiveness in the fight against COVID-19, rare adverse effects of the vaccination have been shown in some studies, including vascular microcirculation disorders and autoimmune and allergic reactions. The biodistribution of mRNA vaccines remains one of the most poorly investigated topics. This mini-review discussed the results of recent experimental studies on humans and rodents regarding the biodistribution of mRNA vaccines, their constituents (mRNA and lipid nanoparticles), and their encoded antigens. We focused on the dynamics of the biodistribution of mRNA vaccine products and on the possibility of crossing the blood–brain and blood–placental barriers as well as transmission to infants through breast milk. In addition, we critically assessed the strengths and weaknesses of the detection methods that have been applied in these articles, whose results’ reliability is becoming a subject of debate.

## 1. Introduction

Faced with the COVID-19 pandemic caused by the SARS-CoV-2 virus, researchers around the world have focused on developing a vaccine as rapidly as possible. Two of the first three SARS-CoV-2 vaccines (mRNA-1273 (Moderna) and BNT162b2 (Pfizer/BioNTech)) were mRNA-based. These vaccines are lipid nanoparticles containing nucleoside-modified RNA that encodes full-length spike protein of SARS-CoV-2. The high degree of protection that these vaccines provide against infection with SARS-CoV-2 and against the development of COVID-19 as well as the low prevalence of adverse effects have proven the high effectiveness and safety of these vaccines [1]. Nonetheless, there are still many gaps in our understanding of the biology of mRNA vaccines formulated with lipid nanoparticles (LNPs).

Vaccination with mRNA–LNPs leads to delayed moderate changes in the level of methylation of CpG islands in peripheral monocytes [2]. Furthermore, some papers have revealed alterations in the transcriptome of peripheral monocytes [3,4,5,6,7]. Recent articles [8,9] have shown that as soon as 1 day after the administration of an mRNA vaccine encoding the SARS-CoV-2 spike protein, S-protein can be detected in the blood of vaccinated people as part of episomes. To date, a lot of evidence has accumulated suggesting that LNPs, mRNA, and protein products of mRNA vaccines are detectable in almost all organs and tissues including the brain, heart, lungs, testes, ovaries, and skin [10,11] and can be associated with the development of neuroinflammation [12]. A rodent study showed that LNPs cause a strong inflammatory response at the injection site and induce inflammation in several tissues, such as the lungs [13]. Additionally, LNPs have been found in breast milk [14] and can be detected in the placenta [15,16]. These properties of mRNA-LNP modalities, on the one hand, ensure their universal applicability to the treatment and prevention of various diseases, and, on the other hand, raise the question of the safety of these modalities. Research into the biodistribution of mRNA therapeutics and their products is still fragmentary. In this review, we summarized the results of recent experimental studies on the detection of RNA, LNPs, and their products in various tissues and organs.

## 2. The Search Strategy

A search for original articles on the biodistribution of mRNA vaccines was performed in the Google Scholar database. We used the following query: (rna vaccine) AND (biodistribution). This search was completed on 1 July 2023, and generated 679 hits; among these hits 377 were review articles. After analyzing all the publications, we determined the criteria for including an article in this review: (1) an experimental study; (2) animals or humans which were vaccinated with LNP-formulated mRNA; (3) the object of the study was the biodistribution of mRNA, LNP, or a protein product of an mRNA therapeutic in vivo or ex vivo. It should be noted that publications investigating biodistribution after immunization of animals with naked mRNA or mRNA delivered with liposomes or peptides were excluded from consideration in this review. An additional search with the following query: (rna vaccine) AND (bio-distribution) was carried out on 5 December 2023, and as a result, 42 new articles were analyzed. As a result, we found 20 articles that complied with the aforementioned criteria. Additionally, we included seven relevant articles that were cited in previously identified articles in this review.

## 3. Biodistribution of an mRNA-Encoded Protein

Reports from preclinical studies by Pfizer/BioNTech and Moderna on the biodistribution of mRNA vaccines and their protein products describe various approaches to assessing the biodistribution by means of laboratory animals (mice, rats, and non-human-like primates) but do not provide exhaustive data [10,17]. To evaluate the distribution of a protein product encoded by mRNA therapeutics, an intravital bioluminescence assay in mice is often used with the help of an in vivo imaging system (IVIS), which allows for the assessment of the distribution of a reporter protein rather than a vaccine antigen. In a report from Pfizer, it was demonstrated [17] that after administration of 2 μg of mRNA within LNPs to BALB/c mice, bioluminescence of firefly luciferase (FLuc) was detectable at various time points at the injection sites (thigh muscles) and in the liver region. Bioluminescence intensity at the injection sites peaked at 6 h after administration of the construct and decreased slowly during the first 72 h; after 6 and 9 days, it diminished to approximately 500 and 1200 times the peak values, respectively. In another study [18], researchers examined the expression profile of Fluc encoded by mRNA as part of LNPs, depending on the dose of this construct. BALB/c mice were injected with luciferase mRNA as part of LNPs at four doses (0.08, 0.4, 2, or 10 μg of RNA), and the protein’s expression was detected using an IVIS at different time points (Table 1). It was found that the intensity of bioluminescence depends on the dose of the administered construct. Maximal bioluminescence intensity was also observed at 6 h after the administration of mRNA–LNPs. Thus, these findings are consistent with the results in the Pfizer report.

Based on the Pfizer report and other studies of biodistribution of reporter proteins, one can conclude that after intramuscular injection of mRNA-LNP, the luminescent signal from luciferase is detected mostly in the muscle (site of injection) and the liver. For example, 3 h after intramuscular injection of 5 μg of mRNA-loaded LNPs to BALB/c mice, robust luciferase expression was found at the injection site and a weak signal was detected in the liver [19]. Similar results were obtained on BALB/c mice, which were injected intramuscularly with 5-10 µg luciferase mRNA-LNP. 6 h after immunization robust luminescence signal was detected by the IVIS method at the site of injection and in the liver [20,21]. Further analysis of ex vivo luminescence in the organs of immunized mice showed the presence of the reporter protein signal in the spleen as well. In the Ripoll et al. study [22] in BALB/c mice, it was also shown that the maximum bioluminescence signal during intramuscular administration of FLuc-LNP occurs at 6 h at the injection site and disappears within 4 days regardless of the composition of the LNP.

There are two reasons for such distribution at the injection site and in the liver. Firstly, the bioluminescence intensity in vivo is dependent on the photon flux per cell, the number of cells, and the migration of photons through tissue [23]. In other words, the larger the organ and the closer it is to the surface, the more pronounced the intensity of bioluminescence will be. Secondly, four-component LNPs composed of ionizable lipids, SM-102 (Moderna) or ALC-0315 (Pfizer/BioNTech/Acuitas), are typically adsorbed to apolipoprotein E (ApoE) in the bloodstream and are taken up by hepatocytes that express high levels of low-density lipoprotein receptors (LDLRs) [24].

Additionally, the kinetics and pattern of biodistribution of the signal of reporter protein significantly depend on the immunization route. It was shown that the duration and intensity of luminescence after the administration of mRNA-LNP in mice decreased in the following order: intradermal > intramuscular > intraperitoneal and subcutaneous > intravenous > intratracheal [25]. Furthermore, high accumulation of the reporter protein occurs in the liver with almost all immunization routes, but in the case of subcutaneous and intradermal routes luciferase expression was observed only at the injection site. Qiu et al. [26] demonstrated that biodistribution depends on the route of administration of a mRNA-based drug, using the example of the administration of luciferase RNA formulated in LNPs to C57Bl/6J mice. Intravenous injection led to the detection of fluorescent proteins in the liver, spleen, lungs, and lymph nodes; the maximum bioluminescence signal occurred 6 h after injection and then slowly decreased. With subcutaneous administration, luciferase expression was observed only in the lymph nodes, the maximum bioluminescence was observed after 24 h and decreased rapidly.

It should be noted that FLuc is a short-lived protein; it has been shown that the half-life of FLuc in a lysate of the HepG2 cell line is ~3 h, and in live cells of the HEK293T line, it is 2 h [27]. The half-life of the spike protein of SARS-CoV-2 is more than 8 h in live BEAS-2B cells [28]. The spike is a transmembrane protein and it can circulate within episomes [9]; hence, FLuc and the spike protein can differ significantly in biodistribution. Thus, the results of studies on the biodistribution of protein products of mRNA therapeutics in rodents are limited to data on reporter proteins, and the number of articles assessing the biodistribution of antigens encoded by vaccines mRNA-1273 or BNT162b2 is rather small.

In human studies, the concentration of a fragment of the spike protein (antigen S1) (the spike protein contains a cleavable S1–S2 site, which enables a release of S1 from the spike trimer [29]) and of the spike protein itself after vaccination with Moderna mRNA-1273 has been measured in plasma using a SIMOA (a single-molecule array) [8]. In a study just cited, measurements were carried out for 29 days after the first vaccination and for 28 days after the second vaccination. The concentrations of the full-length spike protein and its S1 antigen were measured separately because that study tested whether the spike protein is cleaved completely after its synthesis in animal cells. The S1 antigen was detectable in the plasma of 85% of study participants already on the first day after vaccination with the first dose and reached its maximum levels on day 5 (Table 1). The highest concentration of the S1 antigen was 68 ± 21 pg/mL. The concentration became undetectable on day 14. The spike protein was found in 23% of people, and its highest concentrations were observed on the 15th day after the vaccination and were 62 ± 13 pg/mL. The researchers hypothesized that a possible reason is that sometime after vaccination, the immune system eliminates antigen-presenting cells that have the spike protein on the surface, and therefore the full-length protein leaves the cell, continues to circulate in blood plasma, and becomes detectable. In addition, limitations of the SIMOA method itself cannot be ruled out, although its sensitivity is high and ranges from 10 fg/mL to 1 pg/mL [30]. After the second dose of the vaccine, neither S1 nor spike was found, and both antigens remained undetectable until day 56 [8]. In another project [31], which involved the antigen capture ECL (electrochemiluminescent) immunoassay, the spike protein was found in the blood plasma of 96% of vaccinated people on days 1–2 at an average concentration of 47 pg/mL and in 63% on day 7 at an average concentration of 1.7 pg/mL after the first dose of vaccine mRNA-1273 or BNT162b2. After administration of dose 2 of the vaccine, the spike antigen was detectable at low concentrations (on average 1.2 pg/mL) in only 50% of the vaccinated people on days 1–2, and in only one person on day 7.

It is evident that these proteins are poorly detectable or not detectable after vaccination with the second dose; a possible reason is that the analyzed blood plasma samples already contain antibodies to the protein product of the mRNA vaccine, and during the enzyme immunoassay, they compete for binding to the target epitope [31]. In a work by Bansal et al. [9], by means of transmission electron microscopy and antibodies specific to the spike protein, the presence of the spike antigen was registered on the surface of blood plasma exosomes. Additionally, plasma samples from vaccinated healthy individuals were analyzed by western blotting for exosomes carrying the spike antigen. The spike protein was not detectable at 7 days after the first dose of the vaccine but was found at 14 days after the first dose. Two weeks after the second dose, a significant increase in spike protein concentration was observed as compared to previous time points. The amount of the spike antigen declined significantly at 4 months after the double vaccination but was still detectable. That study [9] also revealed that the kinetics of the formation of antibodies to the spike antigen completely correlate with changes in the spike protein amount on the surface of exosomes, that is, the highest concentrations of the antibodies in the blood were found on the 14th day after the vaccination with dose 2 of the construct. The western blot method [32,33] has sensitivity similar to that of the enzyme-linked immunosorbent assay; therefore, it remains unclear why the spike protein was not detectable in the first week after the vaccination. One possible explanation is the use of antibodies specific to different epitopes, as in the case of S1 and spike protein in ref. [8]; besides, the sensitivity of western blot analysis may be lower than that of SIMOA. Of note, during western blotting, denaturing conditions are employed, and this feature prevents competitive binding of blood plasma antibodies and ensures a more accurate determination of the spike antigen concentration.

Research on the biodistribution of spike protein via more invasive approaches has allowed us to evaluate its biodistribution in axillary lymph nodes and human skin [11,31]. A study has qualitatively assessed the presence of the spike antigen in ipsilateral axillary lymph nodes for 7–60 days after full vaccination with mRNA-1273 or BNT162b2 [31]. Immunohistochemical staining for the spike antigen in the lymph nodes of vaccinated patients revealed peak amounts of the spike protein in germinal centers 16 days after dose 2, with the spike antigen still detectable on day 60. At 1–7 days after the vaccination with mRNA-1273 or BNT162b2, the spike protein was found in cutaneous microvasculature in 10 of 12 cases by immunohistochemical staining. This antigen was located in the endothelium of capillaries in the reticular dermis and subcutaneous fat, but vessels containing the spike protein were rare in the tissue sections (less than 5 per sample). Skin samples collected at different time points did not differ either in the amount of the spike protein in the vascular endothelium or in the number of vessels containing this protein.

Thus, studies on humans show the detection of the antigen encoded by mRNA vaccines in blood plasma, lymph nodes, and skin at various time points. The results in most articles on the biodistribution of the spike protein encoded by mRNA vaccines after vaccination in humans indicate that such proteins in blood plasma are detectable as soon as 1–2 days after vaccination with the first dose, meaning that translation can begin almost immediately after vaccine mRNA enters the cell. Data on maximal plasma concentrations are inconsistent and do not correlate well with findings about maximal concentrations of FLuc in animals. After the second dose of the vaccine, the highest concentrations of the spike protein in blood plasma have been observed on days 14–15, and in small quantities, it persists for up to 120 days [9]. The extended presence of the spike protein in the blood of vaccinated people was also confirmed via mass spectrometric analysis [34]. It is noteworthy that in this study, spike protein was still detected in human blood on the 187th day after vaccination.

**Table 1 biomedicines-12-00059-t001:** The list of articles on the biodistribution of antigens encoded by mRNA–LNP vaccines.

mRNA-LNP	Immunization Details	Species	Results (Antigens)	Ref.
mRNA(FLuc)-LNP	A single intramuscular injection of 2 µg of mRNA	BALB/c mice	FLuc was detected at the injection site and liver (max concentration at 6 h)	[17]
mRNA(FLuc)-LNP	A single intramuscular injection of 0.08, 0.4, 2.0, or 10 µg of mRNA	BALB/c mice	FLuc was detected at the injection site and liver (max concentration at 6 h)	[18]
mRNA(Luc)-LNP	A single intramuscular injection of 5 µg of mRNA	BALB/c mice	+++ injection site + liver (3 h after administration)	[19]
mRNA(Luc)-LNP	A single intramuscular injection of 10 µg of mRNA	BALB/c mice	+++ injection site ++ liver + spleen (6 h after administration)	[20]
mRNA(FLuc)-LNP	A single intramuscular injection of 5 µg of mRNA	BALB/c mice	+ injection site (max concentration at 6 h)	[22]
mRNA(Luc)-LNP	A single IV	C57Bl/6J mice	+++ liver ++ spleen + lungs, lymph nodes (max concentration at 6 h)	[26]
	or SC injection		+ lymph nodes (max concentration at 24 h)	
mRNA-1273	13 individuals (first/second dose)	Humans	S1 antigen: plasma (max concentration at 5 days) Spike protein: plasma (max concentration at 14 days)	[8]
mRNA-1273 or BNT162b2	Seven individuals (first/second dose)	Humans	Spike protein: ipsilateral axillary lymph nodes (max concentration at 16 days)	[31]
mRNA-1273 or BNT162b2	~300 individuals (first/second dose)	Humans	Spike protein: plasma (max concentration at 2 days)	[31]
BNT162b2	Eight individuals (first/second dose)	Humans	Spike protein: plasma (max concentration at 14 days)	[9]
mRNA-1273 or BNT162b2	20 individuals (first/second dose)	Humans	Spike protein: plasma (in 50% of subjects, up to 187 days after vaccination)	[34]
mRNA-1273 or BNT162b2	22 individuals (first/second dose)	Humans	Spike protein: skin	[11]

“+”: detectable, the number of plus signs indicates relative levels among different organs and tissues.

## 4. Biodistribution of Vaccine mRNA and LNPs

To investigate the biodistribution of lipids from the LNPs of mRNA vaccines, [^3^H]labeled LNPs as part of an mRNA vaccine (dose: 50 µg/animal) have been used in rats [17]. The radiolabeling method is considered more sensitive than a bioluminescence-based assay and gives a more accurate picture of biodistribution. One paper found that within 48 h, the LNPs spread from the injection site (thigh muscle) to most tissues. Despite the widespread distribution of LNPs, the signal in many tissues was weak, indicating that the distribution of LNPs throughout the rat body was inhomogeneous. Although the ^3^H isotope was found in most tissues within 15 min after the intramuscular administration of the construct, the main target tissues of the mRNA vaccine were the injection site and liver. The highest concentration of ^3^H was registered at the injection site at each time point in rats of both sexes. Maximal plasma concentrations were reached at 1–4 h after the vaccine administration. At 48 h after the injection of the construct, its distribution predominantly involved the liver, adrenal glands, spleen, and ovaries, and the highest concentrations in these organs were observed at 8–48 h after the injection. After 48 h, 21.5% of the total administered ^3^H amount was registered in the liver and much smaller amounts in the spleen (<1.1%), adrenal glands (<0.1%), and ovaries (<0.1%). Concentrations of the detected ^3^H isotope and its tissue distribution patterns were similar between female and male rats. Consequently, the biodistribution of the mRNA vaccine was not sex-specific.

In preclinical studies by Moderna [10], the biodistribution of RNA was evaluated by means of mRNA-1647 after a single intramuscular injection of 100 μg of RNA formulated with LNPs in Sprague–Dawley rats. mRNA-1647 (six mRNAs + LNPs) is a vaccine that includes six different mRNAs of CMV origin encoding full-length glycoprotein B and five other proteins: glycoprotein H (gH), glycoprotein L (gL), UL128, UL130, and UL131A, constituting a pentameric glycoprotein complex. These six mRNAs were encapsulated into LNPs in a mass ratio of 1:1:1:1:1:1. It is believed that the distribution of the mRNA vaccine is determined by LNP composition, whereas the influence of the mRNA itself is thought to be very limited. Therefore, it was admissible that the aforementioned biodistribution study was performed with the same LNPs containing a different mRNA (i.e., mRNA-1647). Indeed, the targeting of delivery is determined by LNPs [35]; however, the results of the quantitative analysis of heterologous RNA were determined by the stability of the mRNA molecule in the cytoplasm [36]. Thus, the biodistribution pattern of mRNA from the actual vaccine mRNA-1273 may be different.

A qualified multiplex branched DNA (bDNA) assay was utilized to assess the biodistribution of mRNA-1647 in various rat tissues [10]. A multiplex bDNA assay is a multiplex technique for the quantitative analysis of mRNA in cell lysates and tissue homogenates and does not require purification of mRNA beforehand [37]. The lower limit of quantitation varied from 0.01 to 0.05 ng/mL depending on the mRNA being tested. Tissues were sampled at 2, 8, 24, 48, 72, and 120 h after administration of the construct (5 rats per group). Quantitation of this mRNA was carried out in the blood and several other organs/tissues. In most tissues, this mRNA was detectable already at the first time point (2 h after injection), indicating rapid and widespread distribution of the vaccine throughout the body (Table 2). Nonetheless, among tissues, there were notable differences in the concentration and in the dynamics of the detection of this mRNA. The highest concentration of this mRNA was recorded at the injection site (muscle), lymph nodes, the spleen, eyes, and the liver. These organs could be regarded as leaders in the level of vaccine mRNA because the highest concentrations of this mRNA were observed in them, and a maximum occurred in the period of 2–24 h after the injection. Another important aspect is the elimination half-life (t_1/2_) of exogenous mRNA in various tissues. Here we can highlight lymph nodes (axillary/popliteal) and the spleen, where elimination half-life was 31.1/34.8 and 63 h, respectively. This finding indicates the long-lasting presence of vaccine mRNA in these organs and may be explained by their important role in the immune system. The shortest t_1/2_ was noted in the blood (2.7–3.8 h), and a day after the injection of mRNA, it was not detectable in the blood at all. At the injection site, despite the high accumulation of this mRNA, the elimination half-life proved to be relatively short: ~15 h. The heart, lungs, testes, and brain were the organs with the least amount of mRNA-1647. Nevertheless, it is worth noting that this mRNA was detectable in the brain with an elimination half-life of 25 h. This result means that mRNA-containing LNPs cross the blood–brain barrier and are present in the central nervous system.

Biodistribution analyses of an mRNA vaccine have also been performed using an influenza vaccine that contains mRNA encoding hemagglutinin [18]. BALB/c mice in that work were administered the vaccine intramuscularly or subcutaneously; next, by the Quantigene 2.0 bDNA Assay, the concentration of this mRNA in various tissues was measured at 2, 4, 8, 24, 48, 72, 96, 120, 168, and 264 h after the injection. After intramuscular injection of the vaccine mRNA, the highest concentration was registered at the injection site and was 5680 ng/mL with an elimination half-life of 18.8 h. In proximal lymph nodes, the maximal concentration (observed in hour 8 after the vaccination) was 2120 ng/mL with t_1/2_ of 25.4 h. In the spleen and liver, the mean peak concentrations were 86.9 and 47.2 ng/mL, respectively. In other tissues and plasma, the level of this mRNA was 100–1000 times lower. It is noteworthy that the mRNA passed the blood–brain barrier, although the maximal concentration of this mRNA in the brain was the lowest (0.429 ng/mL) and was reached at 8 h after the vaccination.

After subcutaneous administration of the mRNA vaccine in the same paper, the maximal concentration of this mRNA at the injection site was 18.2 mg/mL with an elimination half-life of 23.4 h [18]. Among all other tissues, the highest concentration was registered in the spleen (1.66 ng/mL with t_1/2_ of 65.74 h). Only small amounts of this mRNA were found in the heart, kidneys, liver, and lungs 24 h after the vaccination. The biodistribution patterns of vaccine mRNA were generally similar between the intramuscular and subcutaneous administration of the vaccine. Maximal concentrations in both cases were observed at the injection site on the first day after vaccination, although the elimination half-life for the subcutaneous administration of the vaccine was ~5 h longer. Altogether, data on the biodistribution of vaccine mRNA in rodents suggest that an mRNA vaccine spreads from the injection site through the lymphatic system into the systemic circulation and reaches maximal concentration in tissues and organs at 2–8 h after intramuscular administration or at 4–24 h after subcutaneous administration, overall well consistently with data obtained in biodistribution studies of luciferase mRNA-containing LNPs.

In another study, a biodistribution of E6- or E7-encoding mRNA mixed with (human) TriMix and formulated in LNPs was studied [38]. E6 и E7 are the viral oncoproteins, and TriMix is a mix of three mRNAs encoding CD70, CD40L, and a constitutively active TLR4, which increases the strength of the T cell response and is used as an encoded adjuvant [39]. Biodistribution of E6- or E7-encoding mRNA was evaluated in non-human primates (Cynomolgus monkeys) by reverse-transcription quantitative PCR method (RT-qPCR). Animals were immunized twice with a 1-week interval by intravenous injection of 100 μg mRNA in total. 24 h after the second immunization tissues were collected to quantify mRNA content. The highest concentrations of E6 and E7 mRNA were detected in the spleen followed by liver, bone marrow, draining lymph nodes, and lungs. E6- and E7-mRNA levels in kidneys, brain, heart, and adrenal glands were below quantifiable levels.

Studies have been conducted on the biodistribution of vaccine mRNA on Cynomolgus macaques using positron emission tomography-computed tomography (PET-CT) imaging, which allows quantifying the presence of vaccine mRNA in various organs in vivo [40]. In this work mRNA was labeled with an orthogonal dual PET–near-infrared (IR) probe and injected intramuscularly into animals at 200 µg. Detection was carried out 4 and 28 h after administration of mRNA. Fluorescent-labeled mRNA was found at the site of injection in the right quadriceps muscle, as well as in the inguinal, iliac, and para-aortic drainage lymph nodes. 4 h after immunization the fluorescence intensity increased spreading from the injection site to distal lymph nodes. 28 h after immunization the fluorescence intensity strengthened by an average of 70%, but the distribution pattern remained the same. Thus, mRNA reaches draining lymph nodes quickly and continues to accumulate in the body for 28 h after injection.

In human studies, blood samples from 16 people vaccinated with BNT162b2 (the first, second, or boost dose) have been analyzed by RT-qPCR for this mRNA in plasma for up to 27 days after the vaccination (Table 2) [41]. Within hours of the vaccination, this mRNA was found in the blood of the vaccinated and remained detectable in samples taken at 6 and 15 days but was below the limit of quantitation at 27 days. In another study, RNA from a vaccine was found in human blood on the 28th day after vaccination [42]. These results show long clearance times in plasma as compared to the estimates presented by Moderna, who reported that the elimination half-life for blood plasma was much faster, 2.7–3.8 h, and after one day, this mRNA was not detectable at all. On the other hand, it is worth noting that the pharmacokinetic experiments described in the Moderna report were conducted in rats. This fact may explain the substantial difference in the elimination half-life of vaccine mRNA in plasma between the studies. Experiments on rodents typically involve a smaller number of samples taken at different time points and smaller plasma sample volumes as compared to larger species such as humans, even though the doses of mRNA vaccines for rodents and humans are similar. Consequently, larger species are likely to exhibit significantly longer t_1/2_ due to the higher analytical sensitivity [43].

One study involved needle biopsies of axillary lymph nodes in people who received two doses of mRNA-1273 or BNT162b2 and revealed via in situ hybridization the presence of vaccine mRNA in germinal centers of the lymph nodes on days 7, 16, and 37 after the second dose of the vaccine. Rare foci of vaccine mRNA signals were recorded even on day 60 [31]. It was not possible to quantify the vaccine mRNA in that work because the research methodology was qualitative.

In a recent study [44], an autopsy of some tissues was performed in people who died 1–154 days after the last vaccination with mRNA-1273 or BNT162b2. The amount of vaccine mRNA was measured using RT-qPCR in the obtained tissue samples of lymph nodes, spleen, liver, and ventricles. mRNA was found in 73% of axillary lymph node samples of patients who died within 30 days after vaccination, in the amount of 2–7 copies of the target mRNA per 1 ng of total RNA. However, mRNA was not detected in any samples of the spleen, liver, and mediastinal lymph nodes from patients who died more than 30 days after vaccination, although in studies on rodents, mRNA in the liver is always detected. Researchers link this to the fact that doses of the vaccine in rodents in these studies were significantly higher in relation to body weight than doses used in humans. In addition, in 3 out of 13 patients vaccinated with BNT162b2, mRNA was detected in the ventricles in the amount of 2–3 copies/ng of total RNA. For these patients an autopsy was performed within 20 days after the last vaccination, nonetheless, it should be mentioned that all of them had healing myocardial injuries, which could affect tissue permeability.

**Table 2 biomedicines-12-00059-t002:** The list of articles on the biodistribution of vaccine mRNA.

mRNA-LNP	Immunization Details	Species	Results (mRNA)	Ref.
mRNA-1647	A single intramuscular injection of 100 µg of mRNA-1647	Sprague–Dawley rats (only males)	+++ injection site, lymph nodes, liver, spleen, eyes ++ blood + heart, lungs, testes, and brain − kidneys (max concentrations at 24 h)	[10]
mRNA vaccine against Influenza Viruses	A single injection of 6 mg of formulated mRNA, either intramuscularly or intradermally	CD-1 mice (only males)	++++ injection site +++ lymph nodes ++ spleen, liver + blood plasma and other tissues (max concentrations at 24 h)	[37]
E6- or E7-encoding mRNA (mixed with TriMix)	Two doses intravenously (1 week apart) of 100 µg of mRNA in total	non-human primates	+++ spleen ++ liver, bone marrow + lung, lymph node − heart, brain, kidney, adrenal gland (after the 2nd administration)	[38]
mRNA vaccine against yellow fever	A single intramuscular injection of 200 µg	Cynomolgus monkeys	+++ injection site ++ inguinal LNs + iliac and para-aortic LNs (up to 28 h after vaccination)	[40]
BNT162b2	16 individuals (first/second/boost dose)	Humans	+ blood plasma (up to 15 days after vaccination)	[41]
BNT162b2 or mRNA 1273	HCV-positive patients (first/second dose)	Humans	+ blood plasma (up to 28 days after vaccination)	[42]
BNT162b2 or mRNA 1273	Two doses of one of the vaccines	Humans	+ lymph nodes (up to 60 days after vaccination)	[31]
BNT162b2 or mRNA 1273	20 individuals (first/second/boost dose)	Humans	++ axillary lymph nodes + heart − spleen, liver (up to 26 days after vaccination)	[44]
BNT162b2 or mRNA 1273	Healthy lactating individuals (first/second dose)	Humans	+ breast milk (up to 48 h after vaccination, only in ~50% of subjects)	[45,46]
BNT162b2 or mRNA 1273	Healthy lactating individuals (first/second dose)	Humans	− breast milk	[47]
BNT162b2	Healthy lactating individuals (first/second dose)	Humans	+ breast milk (up to 3 days after vaccination, only in ~10% of subjects)	[48]
rabies saRNA vaccine	A single intramuscular or intradermal injection at 0.15 μg	BALB/c mice	+ injection site (max concentration at 4 h)	[49]
saRNA vaccine	A single intramuscular injection	C57Bl/6J mice	+++ injection site, lymph nodes ++ liver, spleen + lungs (up to 21 days after vaccination)	[50]
rabies saRNA vaccine	A single intramuscular injection at 15 µg	Sprague–Dawley rats	+++ injection site, lymph nodes ++ liver, spleen, lungs + blood plasma – brain, kidneys, heart, reproductive organs	[51]
saRNA vaccine against SARS-CoV-2	A single intramuscular injection at 6 µg	Sprague–Dawley rats	+++ injection site, lymph nodes, spleen ++ liver, blood + heart, lungs, kidneys, reproductive organs − brain	[52]
saRNA vaccine against SARS-CoV-2	A single oral administration at 10 µg	BALB/c mice	++ small intestine + large intestine, liver	[53]

“+”: detectable, “−”: not detectable, the number of plus signs indicates relative levels among different organs and tissues.

## 5. Can Vaccine mRNA Cross Blood–Breast Milk and Blood–Placental Barriers?

The study assessing the amount of vaccine mRNA by RT-qPCR in breast milk involved 11 healthy lactating individuals who received either the mRNA-1273 vaccine (*n* = 5) or the BNT162b2 vaccine (*n* = 6) [45]. In 5 out of the 11 women, vaccine mRNA was found in breast milk but no later than 48 h after the vaccination. The concentration of vaccine mRNA among the women varied from 1.3 to 16.78 pg/μL after normalization to starting breast milk volume (mL). It is worth mentioning that in a later work, the same authors quantified vaccine mRNA in the same samples but by a more accurate method: droplet digital PCR [46]. The concentration of vaccine mRNA varied between 1544 and 7605 copies/mL. After 48 h, vaccine mRNA in breast milk was not detectable either, confirming the previous findings. Despite the similarity of the results obtained in the first and second articles, there were discrepancies. If we convert 7605 copies of mRNA per milliliter to pg/mL, taking into account the molar mass of mRNA, then the resulting figure does not exceed 0.1 pg/mL, which actually contradicts the data obtained in the first paper. Another study, which included seven breastfeeding mothers who received either the mRNA-1273 vaccine (*n* = 2) or the BNT162b2 vaccine (*n* = 5), also evaluated the concentration of vaccine mRNA in breast milk by RT-qPCR [47]. The lower limit of detection of the assay was 0.195 pg and 1.5 pg for the BNT162b2 and mRNA-1273 vaccines, respectively. Nevertheless, none of the breast milk samples showed detectable levels of vaccine mRNA. Another study—which included 14 lactating healthcare workers who received two doses of the BNT162b2 vaccine, with the second dose given on day 21—also assessed the concentration of vaccine mRNA in breast milk by RT-qPCR [48]. Thirty-six out of 40 (90%) samples did not have detectable levels of vaccine mRNA. The highest concentration of BNT162b2 mRNA in the tested samples was 2 ng/mL within 1–3 days after the vaccination. Thus, vaccine RNA is indeed capable of crossing the blood–breast milk barrier, but the amounts of detected RNA are low and are eliminated several days after vaccination. These results highlight the need for additional research to better understand this process and its clinical relevance. Nonetheless, these studies are unlikely to refute the importance of vaccinating pregnant and breastfeeding women because vaccines have shown effectiveness at preventing severe types of COVID-19 and protecting both mothers and infants from the infection [54].

In 2023, the first study was conducted to analyze vaccine mRNA in the placenta [16] and to evaluate the ability of mRNA–LNPs to cross the blood–placental barrier [15]. For this purpose, human placental explants were collected during the second and third trimesters of pregnancy, and besides, four cell lines were analyzed: two of placental origin (these are placental choriocarcinoma BeWo cells and choriocarcinoma JEG-3 cells) and two cell lines with epithelial morphology similar to that of the placenta (lung carcinoma A549 cells and embryonic kidney 293T cells). All cell lines and explants were incubated with vaccine mRNA-1273 or BNT162b2 at concentrations of 0.1 or 1 μg/mL for 30 min or 4 h. Next, total RNA was isolated from the cells and explants and analyzed by RT-qPCR for the presence of vaccine mRNA in the cells. Vaccine mRNA in explants was detectable in negligible quantities, but only at the dose of 1 μg/mL of vaccine BNT162b2 (was not detectable at all in the other cases). By contrast, both vaccines were found in BeWo, JEG-3, 293T, and A549 cells in various amounts depending on the dose and time. To more accurately describe the extent of mRNA uptake by human placental tissue, in situ hybridization analysis was additionally performed by means of RNAScope because it is considered more sensitive than RT-qPCR. Under all of the above conditions, vaccine mRNA was not detectable in placental explants. The researchers concluded that the uptake of vaccine mRNA by placental tissue either does not occur or is negligible [16].

In another work, scientists described how they created a treatment for placental insufficiency by means of mRNA formulated with the LNPs that are targeted for mRNA delivery to the placenta [15]. Using an IVIS, they demonstrated in pregnant BALB/c mice that certain formulations of LNPs allowed the therapeutic to penetrate the placenta with high efficiency: 81% of the total luminescent flux. That work also indicated that none of the tested LNPs are capable of crossing the blood–placental barrier because no bioluminescence signals were found in the analyzed fetuses. Therefore, the results of individual studies suggest that vaccine mRNAs are unable to cross the blood–placental barrier; however, to confirm this statement, a larger number of comprehensive studies on different animal species are needed both under steady-state conditions and under conditions of moderate inflammation.

## 6. Biodistribution of a Self-Amplifying mRNA (saRNA) Vaccine

A vaccine based on saRNA that maintains self-replicative activity derived from an RNA virus vector allows for a dose reduction. However, saRNA is relatively large (9000 to 15,000 nt) [55], and hence its biodistribution may have a distinct pattern. The biodistribution of the saRNA vaccine has been studied by Anderluzzi et al. [49]. The investigated vaccine was co-formulated with lipophilic fluorescent dye DiR (1,1′-Dioctadecyl-3,3,3′,3′-Tetramethylindotricarbocyanine Iodide) and injected into BALB/c mice intramuscularly, intradermally or intranasally. Biodistribution of saRNA was detected by IVIS before the vaccine injection and 4, 24, 48, 72, 144 and 240 h after injection. With all routes of administration, the most intense fluorescence signals were found at the injection site. With intramuscular and intradermal delivery, the signal was detected throughout the study with a maximum of fluorescence 4 h after administration. In the case of intranasal administration, the signal was deactivated within 2 days with a maximum of fluorescence 4 h after injection. Perhaps, this is because with nasal administration the systemic absorption of the vaccine was limited: after 4 h fluorescence was detected in the nasal cavity, in the throat and stomach of mice, but not in other tissues.

Kimura et al. carried out a profound analysis of the biodistribution of saRNA-LNP in C57BL/6 mice [44]. Mice were immunized intramuscularly, and 14 h after injection the quantity of saRNA in muscle and liver was measured by RT-qPCR; there was about 4 times more target RNA at the injection site than in the liver. For a more detailed analysis of the biodistribution pattern of the saRNA vaccine, it was co-formulated in LNPs with fluorescent dye XenoLight DiR and then administrated to mice intramuscularly. 24 h after immunization mice were sacrificed, and samples of muscle, draining popliteal, and inguinal lymph nodes (pLN and iLN, respectively), liver, spleen, pancreas, heart, and lungs were collected for IVIS analysis. LNP-DiR signal was found at the site of injection, in draining LNs, liver, spleen, and, to a lesser extent, in lungs, which indicates that saRNA-LNP spread to all major target organs. In the next experiment, authors used reporter mouse lines (LSL-Luc Tg and Ai9 Tg) to determine whether the expression of saRNA-encoded genes would match patterns of LNP distribution. Studies in these mice allow us to detect organs that express Cre recombinase due to Cre/*loxP* recombination since the genome of these mice contains *loxP*-stop-*loxP* cassettes and reporter genes (luciferases and tdTomato, respectively) under the control of the same promoter. Thus, in case of expression of Cre recombinase encoded in saRNA, reporter genes will be expressed in mouse organs. 7 days after intramuscular injection, the expression of reporter genes in both mouse lines was observed only at the injection site; the expression of reporter genes was also observed 21 days after injection, which indirectly indicates the persistence of saRNA at the injection site. In an article on the biodistribution of a novel rabies saRNA vaccine in Sprague–Dawley rats (15 μg RNA), concentrations of saRNA in various organs were assessed by RT-qPCR at 2, 8, 16, 29, and 60 days after vaccination [51]. In several organs (brain, kidneys, heart, ovaries, and testes) this mRNA was not detectable even on day 2 after the immunization. At the injection site (muscle) and in lymph nodes, this mRNA was detectable up to 60 days after the immunization. In the blood, saRNA was found only after 2 days and only in 20% of the vaccinated rats. In the spleen, liver, and lungs, saRNA was present as early as day 2, but mean concentrations did not peak until day 8 or 15. At later time points, mRNA was not found in these organs.

In another work [52], investigators determined the biodistribution of the SARS-CoV-2 saRNA vaccine in Sprague–Dawley rats (6 μg of RNA). On day 2 post-vaccination, this mRNA was registered at relatively high levels at the injection site (muscle) and in lymph nodes and the spleen, and at lower levels in the heart, liver, gonads, lungs, and blood. Levels of this mRNA gradually diminished in all tissues by day 60 but stayed detectable in lymph nodes, the spleen, and muscle. This mRNA was found in the testes only on day 2 and in the ovaries on days 2 and 8. The SARS-CoV-2 saRNA was detectable in the liver of females for longer periods as compared to males. Consequently, studies in rodents demonstrate that saRNA persists in tissues for a longer period. Nonetheless, longer-term research on the presence of saRNAs and their products has not yet been conducted.

In the article by Keikha et al. [53] BALB/c mice aged 6–8 weeks were orally administered with a single dose of 10 µg saRNA encoding S-protein. At two time points (1 and 24 h) mice were sacrificed and samples from the small intestine, large intestine, liver, blood, spleen, and muscle were taken to quantify the S-protein by ELISA. The maximum concentration of spike protein was observed in the small intestine, slightly lower amount was found in the liver and large intestine. This investigation demonstrates that the oral route of administration of mRNA–LNP vaccines is also possible, and in this case, the expression of the immunogen is predominantly observed in intestinal cells, although expression in the liver is still significant.

In summary, the biodistribution of saRNA-based therapeutics does not differ essentially from conventional mRNA-based drugs. Perhaps longer studies are needed to identify differences.

## 7. Concluding Remarks and Prospects

New articles about the biodistribution of mRNA–LNP vaccines and their antigenic products are gradually improving our understanding of the processes involved. It is surprising that although these vaccines have been used in humans for more than 3 years, there is still no complete picture of their biodistribution. Some studies suggest that vaccine components can persist for up to 187 days in the body and penetrate in small quantities through the blood–brain and blood–breast milk barriers but not the blood–placental barrier. Nevertheless, it is too early to draw final conclusions here, and therefore we propose several possible approaches to further research that should fill our knowledge gaps on the biodistribution of mRNA–LNP formulations and the protein products they encode.

First, to elucidate the biodistribution of vaccine antigens in various tissues and organs, it is necessary to conduct comprehensive studies in different animal species: rodents, non-human-like primates, and large animals such as pigs, which share high similarities with humans in body/organ size, anatomy, and physiology. Second, comprehensive postmortem analyses of vaccinated individuals or studies on tissue samples obtained by biopsy should clarify the biodistribution of vaccine components and of encoded protein antigens in humans. Third, improvement of detection methods or use of more accurate assays will prevent the inconsistency of results that is currently observed. Finally, given that biodistribution may depend on the concomitant state of the body (stress, inflammation, comorbidities, and the use of pharmaceuticals), assessment of biodistribution in animal models should be done not only in a steady state but also under various conditions.

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
