# Peer review of "Biodistribution of RNA Vaccines and of Their Products: Evidence from Human and Animal Studies"

_biomedicines, 2023, doi:10.3390/biomedicines12010059_

Round 1

Reviewer 1 Report

Comments and Suggestions for Authors

The mini-review under the title "Biodistribution of RNA Vaccines and of Their Products: Evidence from Human and Animal Studies" submitted to Biomedicines for publication consideration. Though the topic is very interesting, and needs an extensive literature review and more scientific studies. Authors need to improve the manuscript, considering the review article proper rather to revising the previous research and just explaining the results published previously. Like in paragraphs 78–109, studies are explained just to show the previous results; no critical and balanced review is added. While references in text is confusing, for example, line 89 onward is the explanation of one study and only 2 line for Ref23, which makes it difficult to understand. and onward thoughout the manuscript. 

Authors need to add balanced review along with critical thinking, their own experience-based findings, and understandings. Also conclude the overall scenario of the current research and findings of mRNA vaccines rath than SARS. 

Minors include careful revision for English grammar and text structure 

Such as Line 10 to prevent the infectious diseases

Line 11 Name those two vaccine

Line 13 Despite the 

Line 41-43 Repharase the sentence 

Line 44 evidences. 

Line line 69 not the paper it cn be replaced by study, research or findings 

Line 80 delete because 

Line 89 In a study 

.Hopefully these will help to improve review articel.

Comments on the Quality of English Language

Need minor revision for English 

Author Response

Dear Editor and Referees:

Thank you for allowing us to submit a revised draft of our manuscript titled “Biodistribution of RNA Vaccines and of Their Products: Evidence from Human and Animal Studies”. We are thankful to you for reviewing the manuscript and sharing your valuable comments and concerns with us. We have been able to incorporate into the manuscript most of the suggestions provided by the reviewers.

We have highlighted the revisions within the manuscript.

Below, marked in red, are point-by-point responses to the reviewers’ comments and concerns.

Sincerely,

Vasiliy Reshetnikov

Reviewer 1.

The mini-review under the title "Biodistribution of RNA Vaccines and of Their Products: Evidence from Human and Animal Studies" submitted to Biomedicines for publication consideration. Though the topic is very interesting, and needs an extensive literature review and more scientific studies. Authors need to improve the manuscript, considering the review article proper rather to revising the previous research and just explaining the results published previously. Like in paragraphs 78–109, studies are explained just to show the previous results; no critical and balanced review is added. While references in text is confusing, for example, line 89 onward is the explanation of one study and only 2 line for Ref23, which makes it difficult to understand. and onward thoughout the manuscript. 

Authors need to add balanced review along with critical thinking, their own experience-based findings, and understandings. Also conclude the overall scenario of the current research and findings of mRNA vaccines rath than SARS. 

Reply: We appreciate your evaluation. According to this comment, we have added a critical evaluation of results, described the article search methodology, and included new articles published recently.

Minors include careful revision for English grammar and text structure 

Such as Line 10 to prevent the infectious diseases

Line 11 Name those two vaccine

Line 13 Despite the 

Line 41-43 Repharase the sentence 

Line 44 evidences. 

Line line 69 not the paper it cn be replaced by study, research or findings 

Line 80 delete because 

Line 89 In a study 

.Hopefully these will help to improve review articel.

Reply: Thank you, the text has been corrected.

Reviewer 2 Report

Comments and Suggestions for Authors

Pateev et al. present an interesting review on the biodistribution of mRNA vaccines. They cite different studies on humans and nonhuman animals concerning the presence of mRNA, LNPs and encoded protein throughout the body. A special section looks at the permission of the blood-breast milk and bood-placenta barrier.

The work is well founded and includes a wealth of data presented in concise form. The topic of biodistribution of mRNA vaccines is currently under debate, so the manuscript will be of interest to the readers of Biomedicines.

The manuscript is very well written and can be published as submitted.

Author Response

Dear Editor and Referees:

Thank you for allowing us to submit a revised draft of our manuscript titled “Biodistribution of RNA Vaccines and of Their Products: Evidence from Human and Animal Studies”. We are thankful to you for reviewing the manuscript and sharing your valuable comments and concerns with us. We have been able to incorporate into the manuscript most of the suggestions provided by the reviewers.

We have highlighted the revisions within the manuscript.

Below, marked in red, are point-by-point responses to the reviewers’ comments and concerns.

Sincerely,

Vasiliy Reshetnikov

Reviewer 2.

Pateev et al. present an interesting review on the biodistribution of mRNA vaccines. They cite different studies on humans and nonhuman animals concerning the presence of mRNA, LNPs and encoded protein throughout the body. A special section looks at the permission of the blood-breast milk and bood-placenta barrier.

The work is well founded and includes a wealth of data presented in concise form. The topic of biodistribution of mRNA vaccines is currently under debate, so the manuscript will be of interest to the readers of Biomedicines.

The manuscript is very well written and can be published as submitted.

Reply: Thank you, we appreciate your evaluation.

Reviewer 3 Report

Comments and Suggestions for Authors

In this manuscript the authors provide a mini-review of the available literature concerning biodistribution of RNA based vaccines and their products. This is a very important topic and overall, the manuscript is well organized and easy enough to follow. The cited papers are relevant and the authors do a decent job of comparing the results between studies including benefits and drawbacks of the different methods employeed. The reviewers primary concern is whether a mini-review which includes only 38 references (only 15 of which are specific to the subject as indicated in Table 1 and 2) is sufficient to describe the current state of knowledge.

For example, under section 2 regarding biodistribution of mRNA encoded proteins the authors suggest that biodistribution is limited to data on reporter proteins and that data from the two licensed SARS-CoV-2 vaccines is limited. However, in table 1 only two reporter protein papers performed in mice are cited as compared to at least 4 papers on the SARS-CoV-2 vaccines in humans.

The mRNA vaccine technologies used by Moderna and/or Pfizer have been in development for more than 10 years and have included previous clinical trials for other indications (dengue, Zika, etc), so it's hard to believe that there are so few references on the biodistribution of the lipid nanaparticles, the mRNA constructs, or their protein products.

This reviewer suggests that a comprehensive list of the available research on the subject, including search terms, databases, and time frames used should be provided as a supplementary table followed by a justification for limiting this review to the cited papers.

Other items to consider:

Line 129: "not undetectable" doesn't make sense and would suggest that it is detectable. Was the intent to indicate that it is undetectable or not detectable?

Line 245 - 246: "overall well consistently ...." is incomlete. Rewrite for clarity.

Line 256 - 261: does dose administered play into the differences between human and animal data?

Line 305 - 306: semester = trimester?

Comments on the Quality of English Language

Overall the manuscript is easily followed however sentence structure can be improved throughout.

Author Response

Dear Editor and Referees:

Thank you for allowing us to submit a revised draft of our manuscript titled “Biodistribution of RNA Vaccines and of Their Products: Evidence from Human and Animal Studies”. We are thankful to you for reviewing the manuscript and sharing your valuable comments and concerns with us. We have been able to incorporate into the manuscript most of the suggestions provided by the reviewers.

We have highlighted the revisions within the manuscript.

Below, marked in red, are point-by-point responses to the reviewers’ comments and concerns.

Sincerely,

Vasiliy Reshetnikov

Reviewer 3.

In this manuscript the authors provide a mini-review of the available literature concerning biodistribution of RNA based vaccines and their products. This is a very important topic and overall, the manuscript is well organized and easy enough to follow. The cited papers are relevant and the authors do a decent job of comparing the results between studies including benefits and drawbacks of the different methods employeed. The reviewers primary concern is whether a mini-review which includes only 38 references (only 15 of which are specific to the subject as indicated in Table 1 and 2) is sufficient to describe the current state of knowledge.

  1. For example, under section 2 regarding biodistribution of mRNA encoded proteins the authors suggest that biodistribution is limited to data on reporter proteins and that data from the two licensed SARS-CoV-2 vaccines is limited. However, in table 1 only two reporter protein papers performed in mice are cited as compared to at least 4 papers on the SARS-CoV-2 vaccines in humans.

Reply: Thank you for pointing this out. We conducted an additional search for articles citing those already included in the analysis and also conducted an additional search for preclinical reports of vaccines for SARS-Cov2. Experimental results were described and included in Table 1.

  1. The mRNA vaccine technologies used by Moderna and/or Pfizer have been in development for more than 10 years and have included previous clinical trials for other indications (dengue, Zika, etc), so it's hard to believe that there are so few references on the biodistribution of the lipid nanaparticles, the mRNA constructs, or their protein products.
  2. This reviewer suggests that a comprehensive list of the available research on the subject, including search terms, databases, and time frames used should be provided as a supplementary table followed by a justification for limiting this review to the cited papers.

Reply 2-3: According to this comment, we described our search strategy for the experimental articles. In addition, we checked the relevant citations of the retrieved articles. Our method does not conform to PRISMA standards, but we believe that it covered most of the existing studies.

Other items to consider:

Line 129: "not undetectable" doesn't make sense and would suggest that it is detectable. Was the intent to indicate that it is undetectable or not detectable?

Line 245 - 246: "overall well consistently ...." is incomlete. Rewrite for clarity.

Line 256 - 261: does dose administered play into the differences between human and animal data?

Line 305 - 306: semester = trimester?

Reply: Thank you for noticing. The text has been corrected.